# Tight cohesion between glycolipid membranes results from balanced water–headgroup interactions

Matej Kanduč[1,2], Alexander Schlaich[2], Alex H. de Vries[3], Juliette Jouhet[4], Eric Maréchal[4], Bruno Demé[5], Roland R. Netz[2] & Emanuel Schneck[6]

Membrane systems that naturally occur as densely packed membrane stacks contain high amounts of glycolipids whose saccharide headgroups display multiple small electric dipoles in the form of hydroxyl groups. Experimentally, the hydration repulsion between glycolipid membranes is of much shorter range than that between zwitterionic phospholipids whose headgroups are dominated by a single large dipole. Using solvent-explicit molecular dynamics simulations, here we reproduce the experimentally observed, different pressure-versus-distance curves of phospholipid and glycolipid membrane stacks and show that the water uptake into the latter is solely driven by the hydrogen bond balance involved in non-ideal water/sugar mixing. Water structuring effects and lipid configurational perturbations, responsible for the longer-range repulsion between phospholipid membranes, are inoperative for the glycolipids. Our results explain the tight cohesion between glycolipid membranes at their swelling limit, which we here determine by neutron diffraction, and their unique interaction characteristics, which are essential for the biogenesis of photosynthetic membranes.

[1] Soft Matter and Functional Materials, Helmholtz-Zentrum Berlin für Materialien und Energie, Hahn-Meitner-Platz 1, D-14109 Berlin, Germany. [2] Department of Physics, Freie Universität Berlin, Arnimallee 14, D-14195 Berlin, Germany. [3] Groningen Biomolecular Sciences and Biotechnology (GBB) Institute and Zernike Institute for Advanced Materials, University of Groningen, Nijenborgh 7, 9747 AG Groningen, The Netherlands. [4] Laboratoire de Physiologie Cellulaire et Végétale, CNRS, CEA, INRA, Université Grenoble Alpes, CEA Grenoble, 17 rue des Martyrs, F-38000 Grenoble, France. [5] Institut Laue-Langevin, 71 avenue des Martyrs, F-38042 Grenoble Cedex 9, France. [6] Max Planck Institute of Colloids and Interfaces, Am Mühlenberg 1, D-14476 Potsdam, Germany. Correspondence and requests for materials should be addressed to M.K. (email: matej.kanduc@helmholtz-berlin.de) or to E.S. (email: schneck@mpikg.mpg.de).

Amphiphilic lipids are the fundamental building blocks of biological membrane bilayers. Regarding the chemical structure of their hydrophilic headgroup, neutral membrane lipids can be divided into two main classes. The first are lipids with a headgroup chemistry dominated by one large electric dipole (see Fig. 1a), such as the most abundant phospholipid species phosphatidylcholine (PC). The second class involves lipids whose headgroups comprise multiple small electric dipoles, typically polar hydroxyl (OH) groups, such as glycolipids (see Fig. 1b). In nature, membranes in different cell compartments exhibit largely different lipid compositions. Highly dynamic and loosely packed membrane systems, for instance the endoplasmic reticulum or Golgi membranes, which belong to a network of endomembrane compartments all connected via vesicle budding and fusion, are rich in PC lipids[1]. In contrast, structurally more steady and densely packed multilamellar membrane systems, such as myelin sheaths in vertebrates[2] and the photosynthetic membranes (or thylakoids) in plants[3], exhibit high contents in glycolipids displaying multiple OH groups. This correlation suggests an important role of the fundamentally different headgroup architectures illustrated in Fig. 1a,b for the structural and dynamic characteristics of biological membrane systems.

Two uncharged glycolipids, mono- and di-galactosyldiacylglycerol, MGDG and DGDG, respectively, represent more than 80 per cent of the lipids in thylakoid lipid extracts[4]. The total surface area of thylakoid membranes is amplified by hierarchical organization, reaching in certain plants a total area of hundreds of square metres of thylakoids per square metre of leaves[3]. It is particularly striking that MGDG and DGDG are conserved from photosynthetic cyanobacteria to all chloroplasts in eukaryotes, although they are generated by completely different enzymes[5]. To what extent these glycolipids contribute to membrane stack formation and stabilization is under debate[6]. Several studies have shown that the lamellar periodicity of mature thylakoids is governed by membrane proteins[7,8]. However, in both cyanobacterial and eukaryotic thylakoids one finds regions in which the adjacent, glycolipid-rich membranes are in direct close contact and do not accommodate any large proteins[9,10]. Experiments on synthetic glycolipids and natural lipid extracts indicate a significant role of glycolipids in thylakoid membrane stacking: Lipid extracts from spinach chloroplasts, for instance, spontaneously form multilayers[8]. Moreover, DGDG vesicles aggregate, whereas no aggregation is observed for PC lipid vesicles[11]. Consistently, surface force apparatus measurements revealed striking differences in the interaction between pure phospholipids and pure glycolipids (MGDG and DGDG)[12,13]. In a recent study on membrane stacks reconstituted from natural thylakoid lipid extracts[14], it was found that water uptake significantly depends on the lipid composition. Pure DGDG membrane multilayers exhibit the strongest cohesive behaviour, that is, the weakest tendency to take up water from humid air. DGDG but also the other glycolipid mixtures were shown to swell much less than membranes composed of PC lipids[15], which are enriched in the membranes of structurally more dynamic organelles. Even in excess water, the water layer between DGDG membranes remains as thin as $D_w^0 \approx 1.2\,\mathrm{nm}$, as measured in the present work by neutron diffraction (see Methods section), much less than the 2.8–3.3 nm reported for fluid PC lipid membranes[16]. Thus, while membrane proteins play the key functional role in thylakoids, the tight cohesion of thylakoid lipid extract membranes suggests that the lipids by themselves contribute to the tight stacking.

The interaction between membrane surfaces in water was investigated extensively in experimental and theoretical studies[17,18]. It is typically described on a continuum theoretical level by a superposition of electrostatic, dispersion and undulation

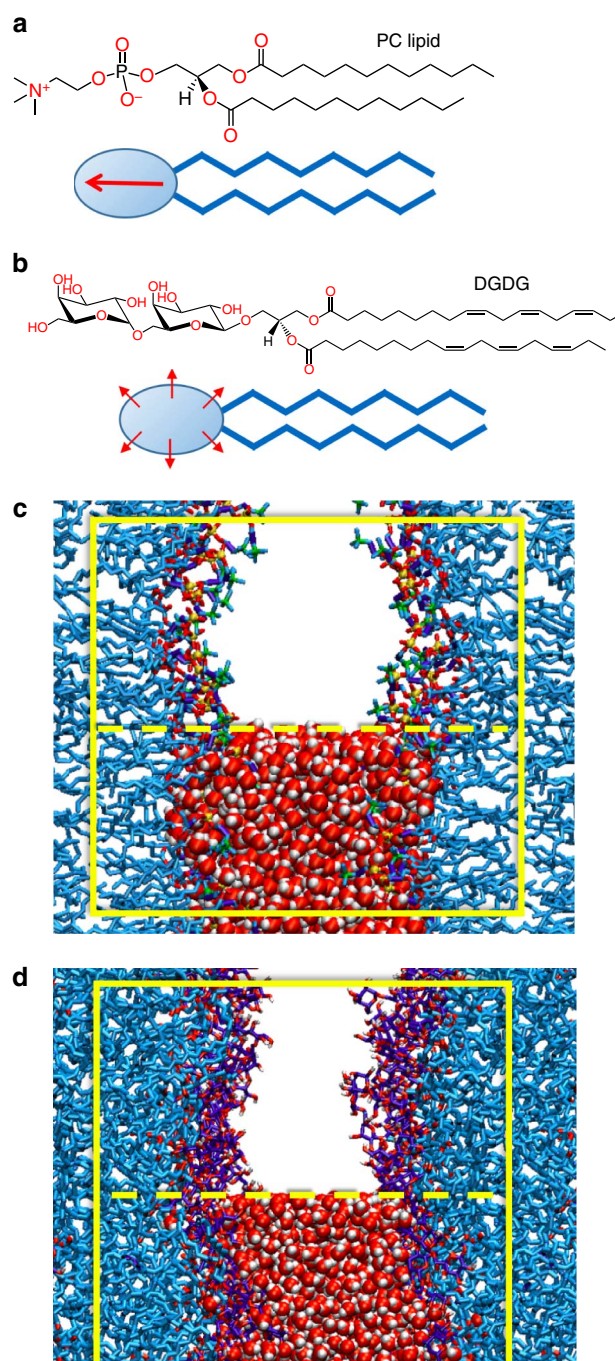

**Figure 1 | Lipid structures and simulation set-up.** Chemical structures of a PC lipid (**a**) and of the glycolipid DGDG (**b**) as representatives of two fundamentally different lipid classes found in nature: Lipids with a headgroup chemistry dominated by one large electric dipole and lipids whose headgroups comprise multiple small electric dipoles in the form of OH groups. Both classes are schematically illustrated below the chemical structures. Dipoles are indicated by arrows. (**c**,**d**) Simulation snapshots of interacting DLPC and DGDG membranes, respectively, both at a large separation of $D_w = 2.3\,\mathrm{nm}$. With periodic boundary conditions in all three directions, the simulations represent a periodic stack of membranes with adjustable hydration level. The simulation boxes are indicated with bright rectangles. For illustration, water molecules are only shown in the lower half of the box.

forces, as well as empirical expressions for the hydration force[19,20]. In one example, Ricoul *et al.*[21] studied the interaction of mixed bilayers of cationic surfactants and glycolipids in aqueous environments. The reduced swelling of glycolipids was explained by an adhesion energy between bilayers varying linearly with the glycolipid molar fraction. However, this level of description does not account for chemical details of the interacting surfaces, which is of great importance at angstrom to nanometre separations. It is therefore now accepted that the molecular structure of surfaces and intervening solvent has to be taken into account explicitly to correctly treat the interaction and to reveal its physical mechanisms on a quantitative level on these length scales[18,22]. Atomistic molecular dynamics (MD) simulations allow the description of biomolecular systems at full chemical detail. However, when it comes to the interaction of extended surfaces such as membranes across an aqueous medium, the explicit treatment of water molecules has made it difficult to work at the correct water chemical potential, the latter being the key control parameter in situations with variable hydration[22]. Over the last few years, we have established methodology to determine the chemical potential of water and the ensuing interaction pressures in atomistic MD simulations with high precision[23,24]. This enables us to investigate membrane interactions on a chemically detailed and mechanistic level.

In the present work, we use atomistic MD simulations to compare the interaction mechanisms of glycolipid and phospholipid membranes. Our simulations quantitatively reproduce the experimentally observed, different pressure-versus-distance curves. Further analysis identifies the hydrogen bond balance as the driving force for the water uptake into the DGDG membranes. The associated repulsion is of short range. Water structuring effects and lipid configurational perturbation, the more long-ranged repulsion mechanisms acting between PC lipid membranes, are found to be irrelevant for DGDG.

## Results

**Area per lipid and pressure–distance curves.** The computer models of the hydrated lipid membranes employ atomistic representations of lipids and water molecules (see Fig. 1c,d for snapshots of PC lipid and DGDG membrane simulations, respectively). With periodic boundary conditions in out-of-plane and in-plane directions, the models represent infinitely extended periodic stacks of membranes in the fluid $L_\alpha$ phase with adjustable hydration level. The details of the model and simulation procedures are described in the Methods section.

Figure 2a shows the average projected area per lipid, $A_l$, of the DGDG membranes as a function of their separation $D_w$ (see Methods section for the definition of $D_w$). In the plot, open squares indicate experimental data by Shipley *et al.*[25] obtained at $T = 293$ K, filled squares indicate our simulation results with semi-isotropic pressure coupling at $T = 300$ K. The obtained quantitative agreement between simulation and experiment is striking. According to the low area thermal expansion coefficient of the DGDG membranes, $\alpha_A = (1.1 \pm 0.3) \times 10^{-3}$ K$^{-1}$ as deduced from the simulations (see Supplementary Fig. 1 and Supplementary Note 1), the temperature difference between the experiments by Shipley *et al.*[25] and our simulations affects the area per lipid by only $\approx 0.006$ nm$^2$, so that it can be safely neglected in this comparison. It is seen that no significant change in $A_l$ occurs as the hydration level is varied. The simulation data are only slightly scattered around the hydration-averaged value $A_l^0 = 0.78$ nm$^2$, indicated in the plot with a horizontal dashed line. This value was used for the production runs with fixed $A_l$. As shown in Supplementary Fig. 2 and described in Supplementary Note 2, variations in the force field parameters qualitatively do

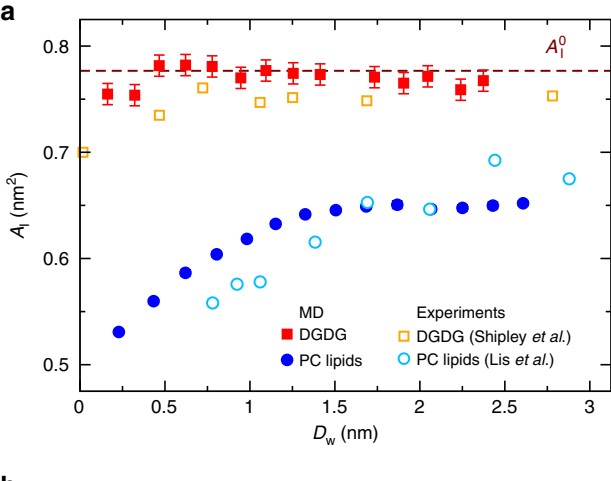

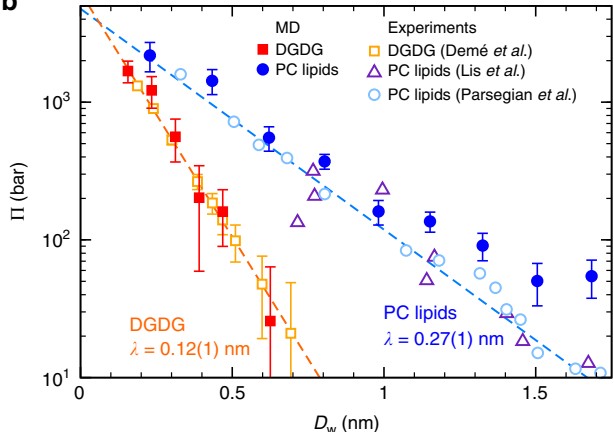

**Figure 2 | Comparison between experiments and simulations. (a)** Area per DGDG and PC lipid headgroup as functions of the water layer thickness $D_w$. Filled symbols indicate simulation results obtained with DGDG and with the PC lipid DLPC with semi-isotropic pressure coupling. Open symbols indicate experimental results for DGDG[25] and DLPC[16] membranes. Error bars for the simulation data represent 1 s.d. of uncertainty and were estimated from the scatter of the points around the plateau value at high hydration. For DLPC, they are smaller than the size of the symbols. The experimental error can be estimated in the same way as $\leq 0.02$ nm$^2$. **(b)** Pressure–distance curves of DGDG[14] and PC lipid membranes[16,28] as obtained in experiments (open symbols) and in the present simulations (filled symbols). Straight dashed lines in the semi-logarithmic plots indicate the best-matching exponential fits to the experimental data points for DGDG[14] and to the combined experimental data sets for egg PC[28] and DLPC[16]. Error bars represent 1 s.d. of uncertainty.

not alter this conclusion. Using area fluctuation analysis in the simulations (see Methods section), the area compressibility modulus of highly hydrated DGDG membranes was determined as $K_A = 0.35 \pm 0.10$ J m$^{-2}$, slightly higher than the values typically reported for PC lipid membranes $K_A = 0.25$ J m$^{-2}$ (ref. 26). In contrast, for PC lipids $A_l$ decreases by $\approx 20\%$ on dehydration to the lowest hydration level investigated (two water molecules per lipid), as shown in Fig. 2a. Open circles indicate experimental data by Lis *et al.*[16] for dilauroyl-PC (DLPC) obtained at $T = 298$ K, filled circles again indicate our simulation results obtained at $T = 300$ K (ref. 27). The good quantitative agreement between experiments and simulations for both DGDG and PC lipid membranes is a first indication of the quality of the employed computer model. The difference in area response of DGDG and PC lipids to the water content already points to distinct hydration mechanisms, as will

be corroborated below. The extent to which water molecules mediate the interaction between saccharide headgroups in the same membrane surface is investigated at high hydration ($n_w = 30$). Each sugar headgroup on average is involved in $2.1 \pm 0.1$ direct hydrogen bonds (HBs, see further text for the definition) with other headgroups in the same surface and in $1.5 \pm 0.1$ 'indirect' HBs, that is, HBs with water molecules that are simultaneously involved in a HB with other headgroups in the same surface. This result indicates that the interactions between the saccharide headgroups are water mediated to a considerable extent.

The swelling of membrane multilayers is commonly described in the form of pressure–distance curves, that is, plots of $\Pi$ versus $D_w$, where $\Pi$ is the so-called equivalent interaction pressure[15,28], which is related to the chemical potential $\mu$ of water:

$$\Pi = -\frac{\mu - \mu_0}{v_w^0}. \tag{1}$$

Here, $\mu_0$ and $v_w^0$ denote the chemical potential and the partial molecular volume, respectively, of pure water in bulk. Note that $\Pi$ is not directly accessed in the experiment. Instead, the actual experimental control parameter is $\mu$ (see Methods section). Figure 2b shows pressure–distance curves of DGDG (squares), together with those determined for PC lipids (circles) for comparison. Open symbols indicate experimental data by Demé et al.[14] ($T = 298$ K), Lis et al.[16] ($T = 298$ K) and Parsegian et al.[28] (room temperature), respectively. For the DGDG data[14], the errors associated with the interaction pressure were estimated by us as explained in the Methods section. For both DGDG and PC lipids, the interaction pressures are positive, that is, repulsive. In other words, work has to be performed to reduce $D_w$. The pressures decay approximately exponentially with increasing $D_w$ and reach magnitudes of several kbars for the lowest $D_w$ studied. The striking difference between DGDG and the PC lipids is the much steeper decay of the repulsive pressure in case of DGDG. An exponential fit to the experimental pressure–distance data yields decay lengths of $\lambda = 0.12 \pm 0.01$ nm and $\lambda = 0.27 \pm 0.01$ nm for DGDG and PC lipids, respectively. This difference manifests itself in the significantly different swelling limit of DGDG and PC lipids in the absence of dehydrating osmotic pressures, $D_w^0$. While $D_w^0$ is as large as $\approx 3$ nm for PC lipids[16], corresponding to $n_w^0 \approx 30$–35 water molecules per lipid, we find a swelling limit of only 1.2 nm for DGDG ($n_w^0 \approx 15$) by neutron diffraction experiments, as described in the Methods section. Moreover, the clear difference in $\lambda$ evidences that the range of the repulsion is not controlled by the properties of water alone, as is often suggested in the literature[29].

The interaction pressures obtained in our simulations (filled symbols in Fig. 2b) via determination of $\mu$ (see Methods section) are seen to be in remarkable quantitative agreement with the experimental data and fully reproduce the difference in the decay length. We remark that, for direct comparison with the experimental data based on equation (1), we translate $\mu$ into $\Pi$ without accounting for the hydration dependence of the partial molecular volume of water (see below). For the water model employed in the simulations, $v_w^0 = 0.030$ nm$^3$ for bulk water at 1 bar and 300 K. The good agreement of our simulations with the experimental data in terms of pressure–distance curves in Fig. 2b demonstrates that the mechanisms through which the hydrated adjacent membranes interact are well captured by the force fields employed in our simulations. In the following, we will analyse the simulation trajectories in detail to rationalize the characteristics of the interaction of DGDG membranes. To this end, we will highlight the differences with the interaction of PC lipid membranes.

**Interaction mechanisms.** The interaction of lipid membranes across a water layer involves a complex interplay of competing molecular interactions that collectively produce a relatively weak net repulsion. In the following, we analyse our simulations such as to identify the dominant repulsion mechanisms, keeping in mind that alternative ways to disentangle the different molecular contributions clearly exist. Figure 3 shows density profiles of water, headgroups and hydrocarbon chains perpendicular to the membrane plane for DGDG (panel a) and PC lipids (panel b)

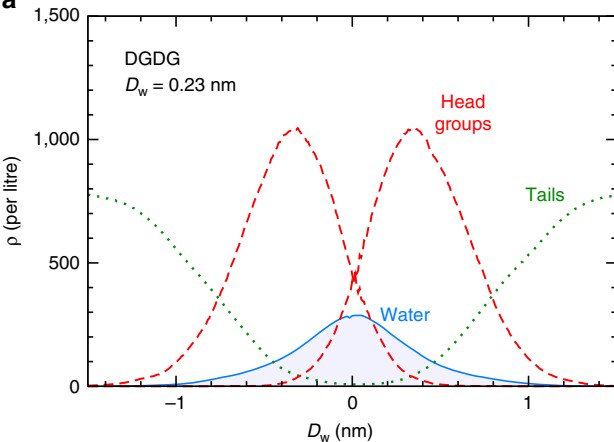
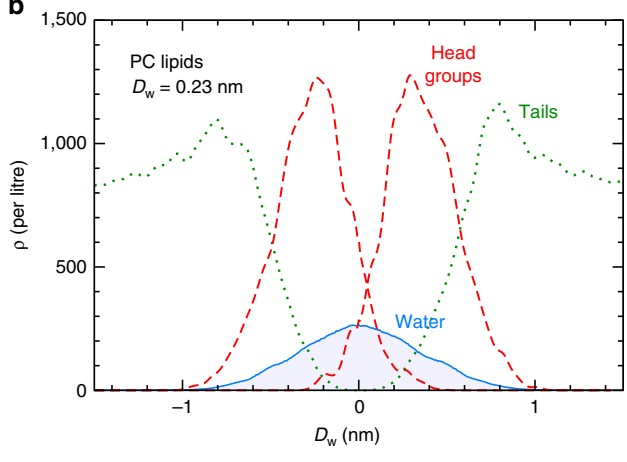
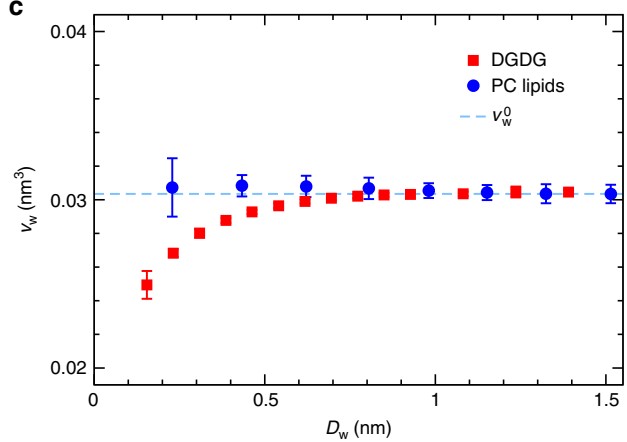

**Figure 3 | Incorporation of the hydration water.** Density profiles of water, headgroups and hydrocarbon chains in hydrated DGDG (**a**) and PC lipid (**b**) membranes at $D_w = 0.23$ nm. (**c**) Partial water volume $v_w$ as a function of $D_w$ for DGDG and PC lipids. Error bars represent 1 s.d. of uncertainty.

at a hydration level representative of significant repulsion, $D_w = 0.23$ nm ($\Pi = 1,200$ bar for DGDG and $\Pi = 2,100$ bar for PC lipids, see Fig. 2b). It can be seen that water molecules and headgroup moieties from both opposing membrane surfaces strongly overlap at the midplane, that is, in the centre of the water distribution. Water uptake will therefore affect lipid–lipid (ll), lipid–water (lw) and water–water (ww) interactions, and the interaction balance will depend on the way additional water molecules are accommodated in the layer of partially hydrated headgroups. Important insight into this can be obtained from the partial water volume, $v_w = (\partial V/\partial N_w)_{p,T}$, defined as the change of the system volume on inserting a water molecule at constant pressure and temperature. In our simulations, we evaluate $v_w$ by fitting the system volume $V(N_w)$ at various hydration levels $N_w$ and taking the derivative with respect to $N_w$. In Fig. 3c, $v_w$ is shown as a function of $D_w$ for DGDG and PC lipids. It can be seen that $v_w$ at large $D_w$ coincides with the bulk water value $v_w^0$ for both DGDG and PC lipids. While for PC lipids, this holds true for the entire hydration range, for DGDG we observe a significant deviation towards lower values, $v_w < v_w^0$, for small $D_w$. This means that on addition of a water molecule at low hydration, the system expands less than a bulk water system does. This observation reflects the non-ideal mixing of water and saccharides. The significant hydration dependence of $v_w$ for DGDG therefore shows that the use of equation (1), assuming $v_w = v_w^0$, is not always justified for the calculation of $\Pi$, neither in experiments nor in simulations. According to the relation between $\Pi$ and the free energy $G$ per area $A$ at constant ambient pressure $p$ and temperature $T$,

$$\Pi = -\frac{1}{A}\left(\frac{dG}{dD_w}\right)_{p,T}, \quad (2)$$

the significant repulsion ($\Pi > 0$) in the short separation range indicates that the incorporation of water into the partially hydrated saccharide layer lowers the free energy. Since both water and saccharides display OH groups at high densities, the key contribution to this free energy reduction on hydration is expected to be related to the formation of HBs. On a simplified level, HBs are electrostatic interactions involving two atoms carrying negative partial charges and a hydrogen atom that is covalently bound to one of them and carries a positive partial charge. To extract HB numbers from our simulation trajectories, we employ the widely used Luzar–Chandler geometric criterion[30], according to which a HB is present if the distance between donor and acceptor atoms is smaller than 0.35 nm and the hydrogen–donor–acceptor angle is smaller than 30°.

The total number of HBs per lipid molecule, $n_{tot}^{HB}$, is defined as the sum of ll, lw and excess ww HBs:

$$n_{tot}^{HB} = n_{ll}^{HB} + n_{lw}^{HB} + n_{ww}^{HB,ex}, \quad (3)$$

where $n_{ww}^{HB,ex} = n_{ww}^{HB} - n_w n_{bw}^{HB}$ and $n_{bw}^{HB} = 1.796$ is the number of HBs per water molecule in bulk water, obtained in separate simulations. Figure 4a shows the change in $n_{tot}^{HB}$ for DGDG on dehydration, $\Delta n_{tot}^{HB}(D_w) = n_{tot}^{HB}(D_w) - n_{tot}^{HB}(\infty)$. It can be seen that $\Delta n_{tot}^{HB}$ increases on swelling, that is, with rising $D_w$, until it reaches zero at large $D_w$. This means that water uptake increases the overall number of HBs. In fact, the decay lengths of $\Pi$ and $n_{tot}^{HB}$ coincide ($\lambda = 0.12$ nm for both, see Figs 2b and 4a where the data are compared with exponential fits), suggesting that hydrogen bonding is at the heart of the swelling mechanism of DGDG membranes. A more rigorous test of this hypothesis is presented in Fig. 4b, where $\Delta n_{tot}^{HB}$ is plotted versus the interaction free energy per lipid $G/N_l$, obtained by integrating the pressure–distance curve, $G = -A\int\Pi dD_w$ according to equation (2). The plot reveals a linear relation between hydrogen bond number $\Delta n_{tot}^{HB}$ and free energy $G$, with zero intercept. The linear fit yields

the slope $d(G/N_l)/d(\Delta n_{tot}^{HB}) = -21 \pm 3$ kJ mol$^{-1}$, which is comparable to experimental estimates of HB free energies[31]. In other words, the repulsion between DGDG membranes can be described entirely in terms of the HB balance. In the inset of Fig. 4a, we further show that with increasing water uptake the number of lw HBs grows at the expense of ll and ww HBs. However, the exponential decay length for all three curves, $\lambda = 0.26 \pm 0.01$ nm, is not reflected in the pressure–distance curve of DGDG (indicated by Fig. 2b), indicating that ll, lw and ww interactions balance each other and thus compensate each other almost perfectly. This is no surprise in view of the similar chemistry of water and saccharide OH groups. For PC lipids, the situation is fundamentally different: the chemistry of the PC headgroup, dominated by a large single dipole of which the positive charge is encased by three hydrophobic methyl groups (see Fig. 1a), has little in common with that of water. In fact, it has been shown that ll, lw and ww interactions for PC lipid membranes are disparate and an energetic preference for lw interactions gives rise to strong repulsion between PC lipid membranes at low hydration[24]. As shown in Fig. 4c, there is no simple proportionality between $G$ and $\Delta n_{tot}^{HB}$ for PC lipids (see Supplementary Fig. 3 and Supplementary Note 3 for the separation dependence of $\Delta n_{tot}^{HB}$). Moreover, in the limit of high hydration (shaded in light blue in Fig. 4c) $G$ varies while the HB number has already saturated ($\Delta n_{tot}^{HB} \approx 0$). This behaviour reflects more long-ranged, non-HB-related repulsion mechanisms between PC lipid membranes at higher hydration, giving rise to the larger decay length in the pressure–distance curves in Fig. 2b.

Two mechanisms are responsible for this long-ranged repulsion: water structuring effects via the strong orientational polarization of the water layers interacting with the lipid headgroups[32], and the configurational entropy of the lipids[33]. In the following, we show that both mechanisms are essentially inoperative for glycolipids like DGDG. Figure 5a presents density profiles of water and lipids in hydrated DGDG and PC lipid membranes at a large separation of $D_w = 2.3$ nm. It is seen that the water density profiles are nearly identical for the two lipid types. In contrast, as shown in Fig. 5b, DGDG and PC lipid membranes lead to very different profiles of water orientation perpendicular to the membrane surfaces, $\langle\cos\theta_w\rangle$ (see inset for the definition of $\theta_w$). For the PC lipids, the water dipoles close to the membrane surfaces at $z = \pm D_w/2 = \pm 1.15$ nm (see vertical dashed lines) are strongly oriented and significant orientation extends virtually all the way to the centre of the water layer (see zoom-in in Supplementary Fig. 4 and Supplementary Note 4), where it has to vanish by symmetry. For DGDG, on the other hand, water orientation is pronounced only in the poorly hydrated inner headgroup regions and insignificant inside the water layer. This dissimilar behaviour can be understood from the different headgroup structures illustrated in Fig. 1a,b: The large and directionally correlated single electric dipoles of PC lipids induce strong water orientation, while the rather isotropically oriented OH groups of the glycolipids do not. Significant repulsion due to water structuring can thus not be expected for DGDG membranes. We move on with Fig. 5c, which shows the angular distributions of DGDG and PC lipid headgroups with respect to the membrane normal for large ($D_w = 2.3$ nm, solid lines) and small ($D_w = 0.6$ nm, dashed lines) separations. Headgroup angles $\theta_l$ are defined by vectors between pairs of headgroup atoms, as illustrated in the figure inset: P–N for PC lipids and opposing headgroup carbons $C_1$–$C_2$ for DGDG. The distributions are weighted with a factor $1/\sin(\theta_l)$, so that a constant distribution would correspond to a random orientation. It can be seen that the angular distribution of the PC headgroups is quite broad for large separations but narrows considerably on dehydration. In fact, more upright conformations with $\theta_l$ smaller

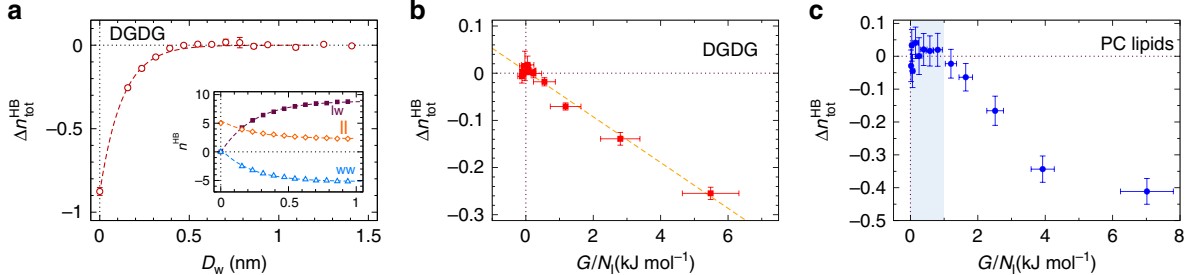

**Figure 4 | Hydrogen bonds.** (**a**) Total number of HBs per DGDG lipid as a function of the membrane separation. The dashed line indicates an exponential fit with decay length $\lambda = 0.12$ nm. Inset: ll, lw and ww HBs. In the plot, ww refers to the excess HB number with respect to bulk water, $n_{ww}^{HB} - n_w n_{bw}^{HB}$, see main text. Dashed lines indicate exponential fits with the same decay length $\lambda = 0.26 \pm 0.01$ nm for all three curves. (**b**) The total number of HBs versus the interaction free energy for DGDG. Dashed line: Linear regression through all data points. (**c**) The same plot for PC lipids. Shaded region: High hydration limit where $\Delta n_{tot}^{HB} \approx 0$. The error bars in all three panels represent 1 s.d. of uncertainty.

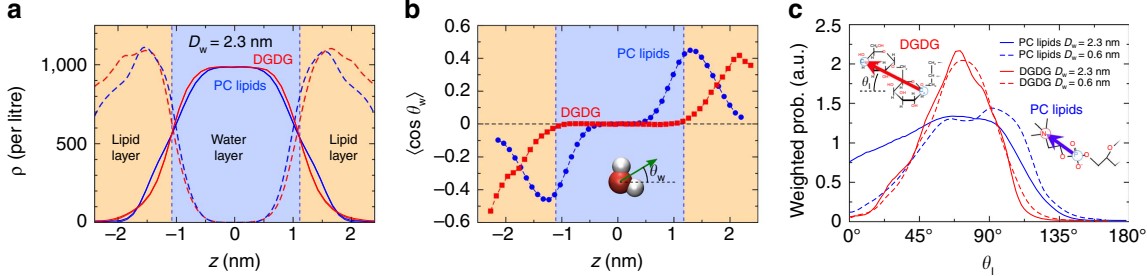

**Figure 5 | Origin of the long-range repulsion.** (**a**) Density profiles of water and lipids in hydrated DGDG and PC lipid membranes at a large separation, $D_w = 2.3$ nm. Vertical lines indicate the membrane surfaces at $\pm D_w/2$. (**b**) Water orientation profiles $\langle \cos \theta_w \rangle$ at the same separation. Inset: Definition of the water dipole angle $\theta_w$. (**c**) Distributions of DGDG and PC lipid headgroup orientations with respect to the membrane normal for large ($D_w = 2.3$ nm, solid lines) and small ($D_w = 0.6$ nm, dashed lines) separations. Inset: Definition of the headgroup vectors.

than approximately 45° are largely suppressed for $D_w = 0.6$ nm. In contrast, the headgroup angular distribution for DGDG is much narrower and, importantly, virtually unaffected by dehydration. (A comparison at the same interaction pressure instead of the same separation yields the same picture, see Supplementary Fig. 5 and Supplementary Note 5.) The comparison in Fig. 5c indicates that entropic repulsion due to a lipid configurational perturbation is of minor relevance for the glycolipid membranes. The latter conclusion is further corroborated by decomposing the interaction free energy $G$ into the enthalpic and entropic contributions, which we show in Supplementary Fig. 6 and discuss in Supplementary Note 6.

## Discussion

With the preceding analysis, we have shown that the water uptake into DGDG membranes is solely driven by the HB balance involved in non-ideal water/sugar mixing. The ensuing repulsion is of much shorter range than that between phospholipid membranes, which exhibit more long-ranged swelling mechanisms. The equilibrium separation between membranes in excess water, $D_w^0$, is known to be governed by the balance between repulsive hydration forces, as quantified in the present study, and the ubiquitous van der Waals (vdW) attraction between the membranes across the water layer[19,20]. More short-ranged repulsion for DGDG thus coincides with smaller $D_w^0$. Because the vdW attractive potential obeys a $D_w^{-2}$ scaling[34], the depth of the adhesive free energy minimum at the equilibrium separation, $G_{adh} \equiv G(D_w^0)$ (which we approximate as the vdW minimum neglecting the tail of the exponential repulsion), scales as $(D_w^0)^{-2}$ and is thus significantly larger for the glycolipids. Comparing DGDG ($D_w^0 = 1.2$ nm, measured by neutron diffraction in the

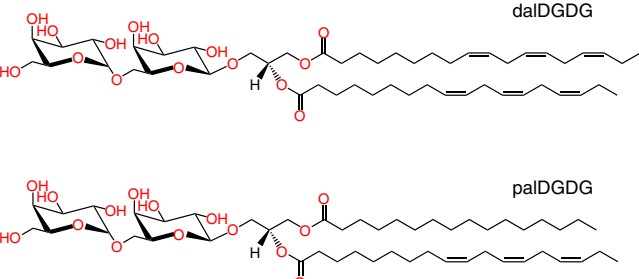

**Figure 6 | Dominant species of DGDG.** Chemical structures of dalDGDG (top) and palDGDG (bottom) lipids, differing in the saturation of the hydrocarbon chains.

present work) and PC lipids ($D_w^0 \approx 3$ nm), we numerically obtain for the ratio of the adhesion strengths $G_{adh}^{DGDG}/G_{adh}^{PC} \propto (D_w^{0,DGDG}/D_w^{0,PC})^{-2} \approx (1.2 \text{ nm}/3.0 \text{ nm})^{-2} \approx 6$, when neglecting differences in the Hamaker constants. With that, our analysis provides the physical explanation for the stronger cohesion of glycolipid membranes and their tendency to form stable lamellar structures. The striking differences in the interaction mechanisms of DGDG and PC lipids, as representatives of the two fundamentally different lipid classes defined in Fig. 1a,b, underline the impact the headgroup design has on membrane interactions. In fact, for certain saccharides, strong attractive contributions to the interfacial force balance have been reported[20,35]. The characteristics of the repulsion between lipid membranes (or hydrophilic surfaces in general) in terms of strength, range and mechanisms are not governed by a universal, water-inherent mechanism, but highly sensitive to the headgroup chemistry.

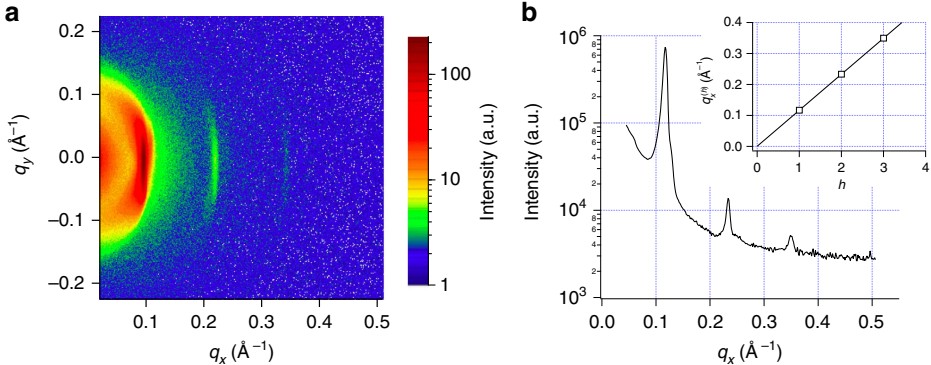

**Figure 7 | Lamellar period at excess hydration.** Neutron diffraction of pure DGDG measured in excess $D_2O$ between two silicon wafers. (**a**) 2D diffraction pattern. (**b**) 1D reduced curve. The insert is a plot of the Bragg reflection ($h=1$ to 3) positions versus $h$. The solid line is a linear fit to the experimental points yielding a period of $54.0 \pm 0.1$ Å.

Clearly, ions are expected to have significant influence on the hydration repulsion, as they affect the balance between effective water–water and water–headgroup interactions, notably HBs[36]. In the future, it would be interesting to compare the influence of ions on the swelling of phospholipids and glycolipids.

The differences in the interaction between glycolipid and phospholipid membranes may be of relevance for protein-free regions in thylakoids, which were reported to exhibit membrane separations in a broad range of 2–4 nm (refs 37,38). In this distance range, phospholipid membranes still exhibit significant hydration repulsion, in contrast to glycolipids, for which the hydration repulsion has basically decayed to zero. This might suggest that protein–protein interactions are not the only factor influencing thylakoid architecture and we speculate that membrane interactions may therefore be relevant even for the evolution of lipid headgroup chemistry.

Complementary information on the tightly packed domains in thylakoids in terms of membrane bending rigidities may be obtained in future studies exploiting the off-specular scattering of neutrons from aligned membrane multilayers[20,39]. Finally, sugar–water interactions are not only relevant for glycolipids but also in a much broader context ranging from sugar solutions[40] and sugar surfactants[41,42] to glycoproteins and sugar-based biomaterials[43]. Some of the concepts presented here may also apply to those problems.

## Methods

**Set up of the computer model.** The computer model of the hydrated glycolipid bilayers employs atomistic representations of lipid and water molecules. Each of the two bilayer leaflets contains 50 DGDG molecules, see Fig. 1b. In the scattering experiments by Demé *et al.*[14], the DGDG sample was dominated by two main DGDG populations with different acyl chains, see Fig. 6: dalDGDG with two threefold-unsaturated C18 chains (18:3) and palDGDG with one 18:3 chain at the sn-2 position and one saturated C16 chain (16:0) at the sn-1 position. In order for the simulations to be comparable to the experiments, dalDGDG and palDGDG were also realized in the computer model and mixed in the approximate ratios reported in the experiments, 80% dalDGDG and 20% palDGDG. The systems were prepared by using the insane.py script[44] available from the Martini website (www.cgmartini.nl). The script can be used to prepare coarse-grained (CG) models of bilayers of varying composition by using lipid templates. After minimization and a short equilibration run at constant area, atomistic details were generated from the CG structures using the 'backward.py' script[45], also available from the Martini website. Mapping files from the CG to the atomistic structures were written for the lipids relevant to these systems. The atomistic structures after backmapping were energy minimized and briefly equilibrated using stochastic dynamics with very short time step[46,47].

**Computer simulations.** All atomistic MD simulations are performed using the GROMACS simulation package[48]. Simulation run files are available as Supplementary Software 1. We use the simple point charge/extended water model[49]. Electrostatics is treated using the particle-mesh-Ewald method[50,51] with a 0.9 nm real-space cutoff. The Lennard-Jones (LJ) interactions are cut off at $r_{LJ} = 0.9$ nm. The simulations are performed with an integration time step of 2 fs in the canonical constant pressure ensemble with periodic boundary conditions. Temperature was maintained at $T = 300$ K, using the Berendsen thermostat[52] with a time constant of 1 ps. The pressure was maintained at $p = 1$ bar using the Berendsen barostat with a time constant of 1 ps.

For the PC lipids, we use the united-atom Berger force field[53]. The bilayer is composed of 72 DLPC molecules (36 in each layer) and subject to semi-isotropic pressure coupling at 1 bar (independent lateral and perpendicular simulation box scaling). For each hydration level, the membranes are equilibrated for at least 5 ns before the production runs, which have durations of 75 ns. The thermodynamic integration (TI) simulations needed for the determination of the water chemical potential (see below) has a total duration of 800 ns per hydration level.

For the glycolipid simulations, we start from the above-described initial configurations. The DGDG force field is based on GROMOS 53a6, which employs united-atom treatment of non-polar hydrogens[54] and is described in refs 46,47. Due to their pronounced hydrogen-bonding capabilities, DGDG bilayers exhibit much longer relaxation times than the PC lipid bilayers. To nevertheless achieve sufficient sampling statistics, additional measures are taken: At first, for each hydration level an initial 80 ns simulation run with semi-isotropic pressure coupling at 1 bar is performed to determine the corresponding equilibrium value of $A_l$. In the next step, however, the lateral box extension, and thus $A_l$, is then kept at the constant value $A_l^0$, as explained in the text. This treatment reduces fluctuations of the box vectors and the hydration water layer thickness and facilitates measurements of the water chemical potential. Second, for each hydration level, 15 independent sets of simulations are averaged, amounting to a total duration per hydration level of 1,000 ns of production runs and 7,500 ns of simulations for TI.

The water layer thickness (or membrane separation) $D_w$ is defined via the number of water molecules per lipid $n_w$, as $D_w = 2 n_w v_w^0 / A_l$, where $v_w^0 = 0.03$ nm$^3$ is the volume of a water molecule in bulk and $A_l$ is the average area per lipid. This definition is commonly used in the experimental literature[28].

The equivalent interaction pressure, computed via equation (1), is in experiments obtained by controlling the water chemical potential $\mu$. In the experimental work on DGDG[14], for instance, $\mu$ is adjusted via the relative humidity $h_{rel}$, as $\mu = \mu_0 + k_B T \ln(h_{rel})$.

We have recently established the methodology to determine the chemical potential of water in atomistic MD simulations with high precision. $\Pi$ is then obtained from the shift in the chemical potential in analogy with equation (1). To determine $\mu = \mu^{id} + \mu^{ex}$, we independently measure its excess and ideal contributions, $\mu^{ex}$ and $\mu^{id} = k_B T \ln(\rho)$, where $\rho$ is the water density. While $\mu$ by definition is constant over the simulation volume in thermal equilibrium, $\mu^{ex}$ and $\mu^{id}$ are not. Due to the inhomogeneous water distribution perpendicular to the membrane surface, $\mu^{ex}(z)$ and $\mu^{id}(z)$ via $\rho(z)$ are functions of the perpendicular coordinate, $z$. As a consequence, $\mu^{ex}$ and $\mu^{id}$ have to be evaluated at the same $z$ position in the simulation box to determine $\mu$. This is facilitated for minimal box vector fluctuations and thus motivates constraining $A_l$ in the production runs. We

## Table 1 | Bragg peak positions.

| Bragg order | $q_x$ (Å$^{-1}$) | $\delta q_x$ (Å$^{-1}$) |
|---|---|---|
| 1 | 0.11697 | 0.000130 |
| 2 | 0.23346 | 0.000103 |
| 3 | 0.34957 | 0.000302 |

then conveniently determine $\mu^{id}$ from $\rho$, whereas for the determination of $\mu^{ex}$ we use a computationally efficient combination of two approaches, the test particle insertion[55] and the TI[56]. Additional methodological details are described elsewhere[57]. At this point, we remark that for consistent evaluations of the LJ energies, the LJ potentials should be shifted by an offset to reach zero at their cutoff distance $r_{LJ}$. The reason is that the particle trajectories in MD are produced by computing the forces from potentials, and therefore any potential that is cut off effectively behaves as being shifted. This fact was ignored in refs 24,27, which led to slightly different pressure–distance curves.

**Area compressibility modulus.** In the simulations with unconstrained lateral box area $A$, the area compressibility modulus $K_A$ is calculated from the fluctuations in the area according to ref. 58

$$K_A = \frac{k_B T \langle A \rangle}{\langle (A - \langle A \rangle)^2 \rangle}. \tag{4}$$

To give an error estimate for $K_A$, the total simulation time is divided into 5–10 blocks (leading to various lengths of 100–200 ns) and the value for $K_A$ is calculated for each block. The analysis was performed for hydration levels of 5 and 20 waters per DGDG lipid and yielded the following results:

$n_w = 5;\ K_A = 0.85 \pm 0.20\ \mathrm{J\,m^{-2}}$
$n_w = 20;\ K_A = 0.35 \pm 0.10\ \mathrm{J\,m^{-2}}$

**Experimental errors of DGDG interaction pressures.** In the original pressure–distance measurements on DGDG membranes by Demé et al.[14], the errors associated with the interaction pressure were not estimated. In their work, the hydrated membrane multilayers were in equilibrium with water vapour of known chemical potential $\mu$, which varies with relative humidity $h_{rel}$ as $\mu = \mu_0 + k_B T \ln h_{rel}$. The equivalent osmotic pressure was then calculated using equation (1), viz.

$$\Pi = -\frac{k_B T}{v_w^0} \ln h_{rel}. \tag{5}$$

The accuracy of the measured osmotic pressure $\Pi$ therefore depends on the accuracy of the measured relative humidity $h_{rel}$. The accuracy of the freshly calibrated humidity sensor can be estimated as $\pm 2$ per cent ($\delta h_{rel} \approx 0.02$). By differentiating equation (5), we obtain the error estimate for the osmotic pressure as $\delta \Pi = (k_B T / v_w^0)(\delta h_{rel}/h_{rel})$. Expressing the relative humidity $h_{rel}$ in terms of the osmotic pressure $\Pi$ using equation (5) then leads to

$$\delta \Pi = \frac{k_B T}{v_w^0} \delta h \exp\left(\frac{v_w^0 \Pi}{k_B T}\right). \tag{6}$$

This expression enables us to estimate the error bars for the DGDG data in Fig. 2b.

**Neutron diffraction experiments.** The lamellar periodicity of DGDG at full hydration is measured by exposing a solid-supported, stacked DGDG sample to excess water (in the form of $D_2O$) between two silicon wafers (Si-Mat, Kaufering, Germany) in a sandwich configuration[39]. Under these conditions, the lamellar stack is at maximum swelling and zero osmotic pressure. The sample has the same composition as the one used in ref. 14 in which the experimental DGDG pressure–distance curve is reported: $\approx 80\%$ dalDGDG and $\approx 20\%$ palDGDG, where dalDGDG has two threefold-unsaturated C18 chains (18:3) and palDGDG has one 18:3 chain at the sn-2 position and one saturated C16 chain (16:0) at the sn-1 position, see Fig. 6.

Neutron diffraction experiments are carried out as described in ref. 14 on the D16 instrument at the ILL Grenoble, France, using a wavelength $\lambda = 4.75$ Å and $\Delta \lambda / \lambda = 0.01$. Data analysis is performed using the ILL in-house LAMP software (www.ill.eu/instruments-support/computing-for-science/cs-software/all-software/lamp): The intensity collected on the 2D detector (Fig. 7a) is reduced to 1D by vertical integration after solid angle and detector pixel efficiency correction. The data are corrected for background by subtraction of the empty chamber signal. The lamellar periodicity ($D$) is determined in a linear fit to three Bragg peak positions (Fig. 7b) versus diffraction order $h$ of the lamellar ($h00$) reflections, according to Bragg's law: $q_x^{(h)} = 2\pi h/D$, where $q_x^{(h)}$ is the magnitude of the scattering vector. Each peak is fitted with a Gaussian function on top of a constant background, yielding the following $q_x$-values and errors ($\delta q_x$) for the peak position (Table 1).

The peak widths are governed by the instrument settings (notably by the collimation of the neutron beam) and do not reflect any disorder of the sample. The linear fit to the peak positions yields $D = 54.0 \pm 0.1$ Å. Assuming a bilayer thickness of 41.7 Å (ref. 25) consistently with ref. 14, this lamellar period corresponds to a bilayer separation distance of 12.3 Å.

**Data availability.** All data supporting the findings of this study are available within the article and its Supplementary Information files.

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

## Acknowledgements

We thank Peter Fratzl for insightful comments and Institut Laue-Langevin for neutron beamtime. E.S. acknowledges financial support by the Max Planck Society and from an Emmy-Noether grant (SCHN 1396/1) of the German Research Foundation (DFG). R.R.N. acknowledges support of the DFG via SFB 1112.

## Author contributions

M.K., R.R.N. and E.S. designed research; M.K. and E.S. performed research; M.K., A.S. and A.H.V. developed analytical tools and carried out the analysis; J.J., E.M. and B.D. contributed experimental measurements; M.K., E.M., B.D., R.R.N. and E.S. wrote the paper.

## Additional information

**Competing interests:** The authors declare no competing financial interests.

**Publisher's note**: 

