## [Peer Review File · Nature Communications]

Reviewers' comments:

Reviewer #1 (Remarks to the Author):

This work is a careful analysis of stacked bilayers of glycolipids, and the free energy variations associated. The most widespread glycolipid present in thylakoid membranes is used as illustration of the general principle proposed. For the efficient photosynthesis membranes do not swell more than 3nm water layer. This mystery was known and already discussed by Pierre Joliot, Vittorio Luzzati in their lectures thirty years ago!

The only experimental equation of state established to my knowledge for mixed glycolipids shows two critical points at each temperature (F. Ricoul et al) as well as the lamellar-micelle osmotic equilibrium. These results could only be very roughly explained semi-quantitatively with lots of contorsions on parameters. The situation is general with glycolipids, starting with the basic observation that sugar-based surfactant micelles are not spherical, even at the cmc, a general observation that is not yet explained theoretically.

To my best knowledge, none of the phase diagram with known osmotic pressure can nowadays not be reproduced experimentally by ANY choice of parameters. The "doxa" standard assumption used in more than 1000 papers in the last ten years (reviewed in a special issue of COCIS five years ago) is that the decay length of the hydration force is constant, and issued of the Marcelja-Radic analytic expression: decay length related to area of contact to volume ratio per water molecule with enthalpic term due to hydrogen bounds with atoms on the other side of the "interface".

Since the the hydration enthalpy of first adsorbed water layer could be measured by calorimetry, "simple " primary hydration force in the absence of antagonistic salts could be easily tested. Most experiments failed to confirm. There was no reason for "cloud points" that complicate the formulation of efficient homecare products based on sugar-based surfactants. Moreover, microemulsions based on sugar-based surfactants were extremely difficult to formulate.

Here, the authors present an extremely convincing argument, based on comparison of phospholipid as a counter-example. The origin: several small dipoles instead of a big dipole in the lipid, is a clear guide of experimentalist and can be easily tested. The differences are clear. So, this paper (one of the most important I had to refer in the last 20 years), not only solves the mystery of the strange result that was obtained in the reference 10, and the huge impact on understanding thylakoid membrane function.

Since the result solves in one step the mystery in a large series of experiments, and the main general concept is very attractive. This it should be cross-checked on all facets; the author here give a magisterial method: compare to the initial work of Le Neveu et al. on other system. There, the methodology proposed here retrieves the known decay in phospholipids. Since this will be in a few years, a ground-lifting seminal paper, the authors moreover perform a parametric study and show stability of the result versus choice of exact value of know parameters. As "sherry on the cake", the examination of the number of hydrogen bond number decay also gives a good hint to the experimentalist, and may also drive to solve the problem of hydration force in the presence of concentrated salt, still an open question, as summarized by S. Marcelja (in his recent work 2011).

1-I am not specialist of MD, but all the techniques seem very carefully picked out and tested. Vast number of papers in coordination chemistry do not even consider grand canonical ensemble, so are mainly producing drawings. Here, the authors even go further with precise evaluation of the entropic term. This is spread in older papers by the same group. I think that in a future "classic" in biophysics, it would be worth of adding a paragraph aimed to non specialist, to re-explain in the case of PL and GL and illustrate with values on DGDG and PL, the results of the methodology developed in refs 17-19. So

a page more for this point on Suppl. Materials would make access to this paper on an even larger community.

2-Final minor point: if the authors have in the data of all MD made for this work some idea of the quadratic free energy per molecule near the thermodynamic area, i.e. something like the cost in DGDG molecule to increase or decrease the area per molecule by 10%,- and therefore to decrease or increase the thickness of the bilayer by 10%? This would be useful to all experimentalists studying details on chain packing in phase diagrams of lipids. If available, this would add a few sentences; or may be I have overlooked that this can be deduced for data already shown.

My two small suggestions of additions are minor, and if authors prefer leave this to further papers, or judge that their previous technical development in refs 17-19 was clear enough not to be illustrated by another example, it does not matter.

In short: extraordinary clear and carefully established paper. General new concept with widespread consequences; Opens solution to several other core open problem in glycolipid self-assembly.

My suggestion would be to publish quickly in present form and advertise specially in press release.

Thomas Zemb

Reviewer #2 (Remarks to the Author):

Reviewer's Comment to Manuscript NCOMMS-16-18679-T

This paper deals with all atomistic MD simulations of glycolipid (dalDGDG and pal DGDG membranes), using GROMACS software. They claimed the simulation results agree well with several sets of previously published results (references 10, 12, and 21), suggesting the tight cohesion. However, despite of the fact that the paper is reasonably well written, the reviewer concludes that this paper is not suited for a broader readership on Nature Communications based on the following reasons.

Major

1) One of my major concerns is that this work does not seem to be biologically relevant. Ample evidence suggested that glycolipids, compared to phospholipids, adjust the membrane-membrane interactions in a different manner. In p. 1, they motivated the readership by pointing out the stacking of thylakoid membranes. However, it is widely known that the membrane-membrane contacts in chloroplasts are highly dynamic. Actually, thylakoid membranes contain a significant fractions of high MW proteins, such as F1F0 ATPase, PHII and PSIII. They are playing vital roles in regulating the dynamics of thylakoid membranes For example, recent article in Scientific Reports (L-R. Stingatu et al. (2015)) used inelastic neutron scattering to unravel how the electron transfer process regulates the dynamic membrane-membrane interaction in cyanobacteria (as depicted in Figure 3), which is, from biological viewpoint, very similar to those in chloroplasts. From this context, I strongly doubt if such an "oversimplified model", though it may appeal some experts from chemical physics, can be accepted as a relevant model of thylakoid.

2) The authors failed to clearly explain how accurately one could compare the experimental results presented in references 10, 12, and 21 and their simulations. For example, I do not understand how accurately one can calculate the area per lipids from small-angle scattering results. It is known that the thermal expansion coefficients of lipids are highly anisotropic and thus the area per lipids strongly

depends on temperature. In Figure 2, there is no error given in Fig 2A. Similarly, no experimental data point in the pressure-distance curve in Fig. 2B shows an error in D_w or P_i (I understand this is osmotic pressure), which seems skeptical. This makes the reviewer extremely suspicious about the accuracy of analyses, such as $\lambda = 0.122 \pm 0.003$ nm (!).

3. The authors presented one single neutron diffraction of DGDG (cannot find if this is a mixture of $d\alpha$ DGDG and $p\alpha$ DGDG) results in Fig. 7. It seems strange, because some of the authors are obviously experts in 2D neutron diffraction, which provides with not only the thickness but also the mechanical properties, as they showed in one of the references (ref. 46). Moreover, the error of periodicity (± 0.1 Å) does not seem reasonable from the peak width in 1D curve, corresponding to the Gaussian roughness. I think it is not appreciated to mix the results from other papers (Fig. 2) with one set of preliminary result (Fig. 7).

To conclude, the reviewer suggests that this paper should be submitted in a more specialized journal, such as *The Journal of Chemical Physics* or *Biophysical Journal*, after adding more deeper 2D neutron diffraction analyses.

Reviewer #3 (Remarks to the Author):

A. The paper reports a computer simulation study of the interaction between two non-charged lipid bilayers with a disugar head-group and the theoretical study is complemented by experimental measurements. Key findings are i) The simulations capture the measured interaction profile with an unprecedented accuracy. ii) Sugar lipids show a clearly more short range repulsion than the corresponding phospholipids (with chains in the liquid state). This has the consequence that there is a stronger attractive minimum for sugar lipids and a larger tendency for bilayer-bilayer cohesion. An affect that the authors argue has physiological relevance. iii) It is conclude that the nature of the bilayer-bilayer repulsion qualitatively depends on the chemical nature of the head-group. This is in contrast to a wide-spread belief the the nature of the repulsion is dependent on a generic property of the solvent water. iv) The repulsion found for the sugar lipids is determined by a short-range hydroxyl-water interaction competing with the direct hydroxyl-hydroxyl interaction between sugar moieties.

B. The simulations of the bilayer-bilayer interactions are of unprecedented accuracy and of a quality clearly beyond what other investigators have accomplished. The authors are able to give a detailed picture of the the molecular interactions responsible for the swelling of the bilayer. This is, in my personal view, this has been accomplished for the swelling of electrically neutral amphiphiles. (The phospholipid case has not yet been satisfactorily analysed).

C. The simulation methodology has been used previously by the authors and it is highly appropriate for the problem considered. The data are in general of high quality and they are discussed in a clear way. Some details of the presentation is discussed under point E.

D The handling of the data is, as far I can judge, fully satisfactory.

E. The main, important conclusions are clearly justified from the data. A few details can be improved. i) The chemical composition of the lipid chains are not clearly specified in the experimental section. ii) It is of some interest to what extent water molecules are mediating the in-plane sugar-sugar interactions. This should be clear fro the simulations. iii) It could help the reader if the authors specify the ration water molecules per head-group at the swelling limit. iv) The authors discuss sugar-water interactions referring to both dipole-dipole interactions and to hydrogen bonds. A typical Nature reader isn't aware of the fact that both are electrostatic in nature and closely related. The reader would be helped by a more clear discussion of this point. v) A related issue is that the authors count hydrogen bonds in a

digital manner. It is the firm opinion of the reviewer that this is a futile exercise in a liquid context (except in some special cases) vi) The molecular description of the sugar-water interaction is valid in a much broader context and relevant for a sugar solutions as well as for glycoproteins. The authors could point this out. vii) The authors discuss the zwitterionic phospholipid case as a comparison and trace the difference in terms of the size of the dipoles giving more water polarisation. In the context they refer to the model by Radic & Marčelja seemingly forgetting that this model starts from the situation where surface dipole - water polarisation is gone.

F. See point E above.

G. The authors seem to have missed that Marra and coworkers have measured interactions between glycolipid bilayers in the mid 1980-ties. (See J. Marra J Colloid Interface Science 1986 and references therein). There is also an extensive literature on the aggregation behaviour of sugar surfactants. These systems show a behaviour in qualitative accordance with the findings of the authors.

H. The paper is overall clearly written. However, the section on interaction mechanism is difficult to read. This reflects the fundamental problem that a simulation is based on complex relatively strong molecular interactions that collectively produce a relatively weak net bilayer-bilayer interaction. There is no unique way to disentangle the different molecular contributions. The authors make a good effort to solve the problem, but a shortening of this section might make it more clear and possibly also more valid.

REVIEWERS' COMMENTS:

Reviewer #1 (Remarks to the Author):

ALL (minor) comments made about this beautiful work that will become a "classic" cornerstone in biophysics have been answered to in an excellent way.

Reviewer #3 (Remarks to the Author):

The manuscript has gained considerably in clarity by the revision and I recommend publication. There are some qualitative arguments in the paper which I personally don't support. However in these matters I regard the paper as a very valuable contribution to an ongoing scientific debate.

Reviewer #1 (Remarks to the Author):

This work is a careful analysis of stacked bilayers of glycolipids, and the free energy variations associated. The most widespread glycolipid present in thylakoid membranes is used as illustration of the general principle proposed. For the efficient photosynthesis membranes do not swell more than 3nm water layer. This mystery was known and already discussed by Pierre Joliot, Vittorio Luzzati in their lectures thirty years ago!

The only experimental equation of state established to my knowledge for mixed glycolipids shows two critical points at each temperature (F. Ricoul et al) as well as the lamellar-micelle osmotic equilibrium. These results could only be very roughly explained semi-quantitatively with lots of contorsions on parameters. The situation is general with glycolipids, starting with the basic observation that sugar-based surfactant micelles are not spherical, even at the cmc, a general observation that is not yet explained theoretically.

To my best knowledge, none of the phase diagram with known osmotic pressure can nowadays not be reproduced experimentally by ANY choice of parameters. The "doxa" standard assumption used in more than 1000 papers in the last ten years (reviewed in a special issue of COCIS five years ago) is that the decay length of the hydration force is constant, and issued of the Marcelja-Radic analytic expression: decay length related to area of contact to volume ratio per water molecule with enthalpic term due to hydrogen bounds with atoms on the other side of the "interface".

Since the the hydration enthalpy of first adsorbed water layer could be measured by calorimetry, "simple " primary hydration force in the absence of antagonistic salts could be easily tested. Most experiments failed to confirm. There was no reason for "cloud points" that complicate the formulation of efficient homecare products based on sugar-based surfactants. Moreover, microemulsions based on sugar-based surfactants were extremely difficult to formulate.

Here, the authors present an extremely convincing argument, based on comparison of phospholipid as a counter-example. The origin: several small dipoles instead of a big dipole in the lipid, is a clear guide of experimentalist and can be easily tested. The differences are clear. So, this paper (one of the most important I had to refer in the last 20 years), not only solves the mystery of the strange result that was obtained in the reference 10, and the huge impact on understanding thylakoid membrane function.

Since the result solves in one step the mystery in a large series of experiments, and the main general concept is very attractive. This it should be cross-checked on all facets; the author here give a magisterial method: compare to the initial work of Le Neveu et al. on other system. There, the methodology proposed here retrieves the known decay in phospholipids. Since this will be in a few years, a ground-lifting seminal paper, the authors moreover perform a parametric study and show stability of the result versus choice of exact value of know parameters. As "sherry on the cake", the examination of the number of hydrogen bond number decay also gives a good hint to the experimentalist, and may also drive to solve the problem of hydration force in the presence of concentrated salt, still an open question, as summarized by S. Marcelja (in his recent work 2011).

We thank the reviewer for this favorable evaluation and for reminding us of the important work of Ricoul et al., which is well suited to illustrate the broader relevance of the phenomenon investigated in our manuscript.

In the Introduction section of the revised manuscript (page 2) we therefore make reference to the work of Ricoul et al.:

"In one example, Ricoul et al. studied the interaction of mixed bilayers of cationic surfactants and glycolipids in aqueous environments [Ricoul et al., 1998]. The reduced swelling of glycolipids was explained by an adhesion energy between bilayers varying linearly with the glycolipid molar fraction."

In addition, we mention in the discussion section of the revised manuscript (page 8) that ions can have a significant influence on the balance between effective water/water and water-headgroup

interactions with consequences for the resulting swelling behavior, and refer to the work of Marcelja in this context:

"Clearly, ions are expected to have significant influence on the hydration repulsion, as they affect the balance between effective water-water and water-headgroup interactions, notably hydrogen bonds [Marcelja, 2011]."

1-I am not specialist of MD, but all the techniques seem very carefully picked out and tested. Vast number of papers in coordination chemistry do not even consider grand canonical ensemble, so are mainly producing drawings. Here, the authors even go further with precise evaluation of the entropic term. This is spread in older papers by the same group. I think that in a future "classic" in biophysics, it would be worth of adding a paragraph aimed to non specialist, to re-explain in the case of PL and GL and illustrate with values on DGDG and PL, the results of the methodology developed in refs 17-19. So a page more for this point on Suppl. Materials would make access to this paper on an even larger community.

As suggested by the reviewer, we have added to the Supporting Material a section in which we decompose the interaction free energy into its enthalpic and entropic contributions and discuss the results obtained for the studied glycolipids and phospholipids. We refer to this section in the main text on page 7 of the revised manuscript (at the end of the Results section):

"The latter conclusion is further corroborated by decomposing the interaction free energy G into the enthalpic and entropic contributions, which we show and discuss in the Supporting Material."

2-Final minor point: if the authors have in the data of all MD made for this work some idea of the quadratic free energy per molecule near the thermodynamic area, i.e. something like the cost in DGDG molecule to increase or decrease the area per molecule by 10%,- and therefore to decrease or increase the thickness of the bilayer by 10%? This would be useful to all experimentalists studying details on chain packing in phase diagrams of lipids. If available, this would add a few sentences; or may be I have overlooked that this can be deduced for data already shown.

As suggested by the reviewer, we have included information on the area compressibility modulus of the glycolipid membranes as obtained in the simulations (page 4 of the revised manuscript and Supporting Material):

"Using area fluctuation analysis in the simulations (see Supporting Material), also the area compressibility modulus of highly-hydrated DGDG membranes was determined as $K_A = 0.35 \pm 0.10$ J/m², slightly higher than the values typically reported for PC lipid membranes $K_A = 0.25$ J/m² [27]."

My two small suggestions of additions are minor, and if authors prefer leave this to further papers, or judge that their previous technical development in refs 17-19 was clear enough not to be illustrated by another example, it does not matter.

In short: extraordinary clear and carefully established paper. General new concept with widespread consequences; Opens solution to several other core open problem in glycolipid self-assembly.

My suggestion would be to publish quickly in present form and advertise specially in press release.

Reviewer #2 (Remarks to the Author):

This paper deals with all atomistic MD simulations of glycolipid (dalDGDG and pal DGDG membranes), using GROMACS software. They claimed the simulation results agree well with several sets of previously published results (references 10, 12, and 21), suggesting the tight cohesion. However, despite of the fact that the paper is reasonably well written, the reviewer concludes that this paper is not suited for a broader readership on Nature Communications based on the following reasons.

Major

1) One of my major concerns is that this work does not seem to be biologically relevant. Ample evidence suggested that glycolipids, compared to phospholipids, adjust the membrane-membrane interactions in a different manner.

As pointed out by the reviewer and as explained in the introduction of our manuscript, there have been several studies indicating differences in the membrane-membrane interactions of phospholipid and glycolipid membranes. In our work we make the first rigorous comparison between experimental data on phospholipids and glycolipids in terms of pressure-distance curves and identify the physical mechanisms underlying this different behaviour, which has far-reaching biological and technological consequences.

In p. 1, they motivated the readership by pointing out the stacking of thylakoid membranes. However, it is widely known that the membrane-membrane contacts in chloroplasts are highly dynamic. Actually, thylakoid membranes contain a significant fractions of high MW proteins, such as F1F0 ATPase, PHII and PSIII. They are playing vital roles in regulating the dynamics of thylakoid membranes. For example, recent article in Scientific Reports (L-R. Stingatu et al. (2015)) used inelastic neutron scattering to unravel how the electron transfer process regulates the dynamic membrane-membrane interaction in cyanobacteria (as depicted in Figure 3), which is, from biological viewpoint, very similar to those in chloroplasts. From this context, I strongly doubt if such an "oversimplified model", though it may appeal some experts from chemical physics, can be accepted as a relevant model of thylakoid.

The biological relevance of our work is in part highlighted by the publication by Stingaciu et al. These authors follow the classical view, in which thylakoids are stacked only because of protein-to-protein interactions involving photosystem antennas, and not because of lipid-lipid interactions. Their paper addresses the existence of inter-thylakoid regions, which are in such close vicinity that they cannot accommodate any large proteins. However, it does not address the forces in these protein-free, closely interacting membrane regions, which according to Dekker and Boekma 2005 exhibit separations of only 2nm (Daum et al., 2010: 2-4 nm).

In our work, we quantify and mechanistically interpret the different interaction forces acting between glycolipid and phospholipid membranes. Our results suggest that these differences have an impact on thylakoid membrane interactions in the protein-free regions. Importantly, the differences result from the lipids themselves and therefore are not a consequence of the proteins. Our work is an example of biophysical research where it is shown that a salient and biologically important feature can be explained by only one component of a complex biological system. Our aim is not to make a model for the biological function of thylakoid membranes; our goal is to explain the role of glycolipids in the tightly connected membrane regions.

Our work implies that the protein-to-protein interactions are not the sole determinant in the stack formation as generally thought. With that, the studied lipid-lipid interactions are not only relevant in the biogenesis of nascent protein-free thylakoids, but they could also be critical for the stability of mature photosynthetic membranes from cyanobacteria to chloroplasts.

In response to the Reviewer's comment and in order to further clarify the biological relevance of our present study, we have included in the revised manuscript the following new sentences:

"[...] in both cyanobacterial and eukaryotic thylakoids one finds regions in which the adjacent, glycolipid-rich membranes are in direct close contact and do not accommodate any large proteins [Nagy et al. 2011, Stingaciu et al. 2016]." (page 2),

"Thus, while membrane proteins play the key functional role in thylakoids, the tight cohesion of thylakoid lipid extract membranes suggests that the lipids by themselves contribute to thylakoid stacking." (page 2), and

"The differences in the interaction between glycolipid and phospholipid membranes are most pronounced in the separation range relevant for the protein-free thylakoid regions in which the lipid membranes are separated only by 2-4 nm [Dekker et al., Daum et al.]. This indicates that protein-protein interactions are not the only factor determining thylakoid stacking and suggests that membrane interactions are relevant for the evolution of lipid headgroup chemistry." (page 8).

Finally, in view of the thylakoid membrane dynamics reported in the work of Stingaciu et al., we removed from the introduction the misleading term "static" when referring to thylakoids. In this way we avoid the wrong impression that thylakoid membranes do not exhibit dynamics.

2) The authors failed to clearly explain how accurately one could compare the experimental results presented in references 10, 12, and 21 and their simulations. For example, I do not understand how accurately one can calculate the area per lipids from small-angle scattering results.

The experimental data shown in Fig. 2A are by Shipley et al. [25] and Lis et al. [16]. In their studies they determined the average area per lipid from the lamellar periodicity measured by diffraction, from the independently measured lipid/water stoichiometry, and from the known volumes of lipid and water molecules. This method can be considered very accurate. Although no experimental error is given in these experimental papers, the statistical error can be estimated as $\leq 0.02 \text{ nm}^2$ from the scatter of the data points in the plateau region at high hydration (this is now explained in the caption of Fig. 2 in the revised manuscript). It is obvious that the reported differences in terms of magnitude and hydration dependence between PC lipids and DGDG are much larger than the measurement errors and our simulation results clearly capture these differences. In the revised manuscript we have also added error bars to the DGDG simulation data in Fig. 2A estimated from the scatter of the data points around the plateau region at high hydration (now explained in the caption of Fig. 2).

It is known that the thermal expansion coefficients of lipids are highly anisotropic and thus the area per lipids strongly depends on temperature.

We thank the reviewer for the remark on the temperature dependence of the average area per lipid. In order to confirm that the minor differences between the simulations temperature and the temperatures in the various experiments (which are reported in the revised manuscript) have no significant effect, we have evaluated the area per lipid of DGDG membranes in simulations at different temperatures. It is found that the area thermal expansion coefficient of the DGDG membranes is only $\alpha_A = (1.1 \pm 0.3) \times 10^{-3} \text{ K}^{-1}$ (see page 3 of the revised manuscript and additional Supporting Material). Temperature differences of few K will therefore only have a negligible effect. This is now mentioned on page 3 of the revised manuscript.

In Figure 2, there is no error given in Fig 2A.

See above.

Similarly, no experimental data point in the pressure-distance curve in Fig. 2B shows an error in D_w or P_i (I understand this is osmotic pressure), which seems skeptical. This makes the reviewer extremely suspicious about the accuracy of analyses, such as $\lambda = 0.122 \pm 0.003$ nm (!).

We agree with the reviewer that it would be desirable to have error bars for the experimental data on DGDG in Fig. 2B. Unfortunately, error bars are not given in reference [Demé et al. 2014] from which the experimental data were taken. In the revised manuscript we have therefore included error bars for Π in Fig. 2B, estimated a posteriori from the accuracy of the humidity sensor (see new section in Supporting Material). The error bars in D_w are of the order of ± 0.1 Å (see the third point further below) and therefore comparable to the symbol size. Including errors in Π and D_w in the exponential fit leads to an improved estimate of the λ -uncertainty: $\lambda = 0.12 \pm 0.01$ nm (page 4 of the revised manuscript). We agree with the reviewer that the original error estimate (± 0.003 nm), which was obtained by fitting the data from reference [Demé et al. 2014] without error bars, was not very meaningful. For the experimental data on PC lipids measured in the 1970s and 1980s (refs. 16 and 26) we consider it unnecessary to reconstruct their error bars. At first, their inherent scatter reflects the statistical uncertainty of each individual data point. Secondly, we do not have the information at hand to reconstruct the error bars. And finally, these classical data have been referred to 100s of times and their decay length of ≈ 0.3 nm has been frequently reproduced.

3. The authors presented one single neutron diffraction of DGDG (cannot find if this is a mixture of daIDGDG and paIDGDG) results in Fig. 7.

The single measurement at excess hydration (Fig. 7) was carried out to complement the previously published experimental results by Demé et al. [14] obtained at controlled hydration. The measured water layer thickness at the swelling limit serves for the interpretation of the present simulation results (see Discussion section). In the revised manuscript, the sample composition used for the diffraction measurements at excess hydration (Fig. 7) is now more clearly described (page 12):

"The sample has the same composition as the one used in Ref. [Demé et al. 2014] in which the experimental DGDG pressure–distance curve is reported: $\approx 80\%$ daIDGDG and $\approx 20\%$ paIDGDG, where daIDGDG has two threefold-unsaturated C18 chains (18:3) and paIDGDG has one 18:3 chain at the sn-2 position and one saturated C16 chain (16:0) at the sn-1 position, see Fig. 6."

It seems strange, because some of the authors are obviously experts in 2D neutron diffraction, which provides with not only the thickness but also the mechanical properties, as they showed in one of the references (ref. 46).

The reviewer rightfully points out that off-specular neutron diffraction can be analyzed to obtain mechanical properties of the interacting bilayers, and we agree that such investigations could be very interesting also for the case of membranes composed of thylakoid lipid extracts. This is now stated in an additional sentence on page 8 of the revised manuscript:

"Complementary information on the tightly packed domains in thylakoids in terms of membrane bending rigidities may be obtained in future studies exploiting the off-specular scattering of neutrons from aligned membrane multilayers [20, 39]."

However, since the present work is not focused on the analysis of membrane mechanical properties, the experimental settings to obtain the neutron diffraction data were not optimized for an off-specular analysis. Moreover, we believe that a full-scale off-specular analysis (as in Refs. 20 and 39) together with the corresponding explanation and description would go far beyond the scope of the present study, which is focused on pressure–distance curves and on their physical interpretation.

Moreover, the error of periodicity ($\pm 0.1 \text{ \AA}$) does not seem reasonable from the peak width in 1D curve, corresponding to the Gaussian roughness.

The lamellar periodicity was determined in a linear fit to three Bragg peak positions. The widths of these peaks are governed by the instrument settings (notably by the collimation of the neutron beam) and do not reflect any roughness or disorder of the sample. The instrumental peak widths do not affect the precision with which the peak center position can be determined. This notion is widely accepted and relied on in the field of x-ray and neutron diffraction. This is now explained on page 12 of the revised manuscript.

Each peak is fitted with a Gaussian function plus background. The fitting procedure, as implemented in the data reduction software "LAMP" of the Institut Laue-Langevin, yields the following values and errors for the peak position on the q-axis (now included in Supplementary Information):

Bragg order	q [\AA^{-1}]	err [\AA^{-1}]
1	0.11697	0.000130
2	0.23346	0.000103
3	0.34957	0.000302

Accounting for the errors, the linear fit yields the slope $b = 0.11639 \pm 0.00014 \text{ \AA}^{-1}$. The lamellar periodicity then follows as $D = 1/b = 53.98 \pm 0.06 \text{ \AA}$, where the error is obtained via Gaussian error propagation. In view of the additional, very small error associated with the neutron wavelength, we decided to round up the error to 0.1 \AA , so that we end up with the result $D = 54.0 \pm 0.1 \text{ \AA}$ as presented in the manuscript. Finally, we remark that, although we believe that 0.1 \AA is a good error estimate, even a precision of $\pm 1 \text{ \AA}$ would be more than sufficient to support the conclusions drawn in the manuscript.

I think it is not appreciated to mix the results from other papers (Fig. 2) with one set of preliminary result (Fig. 7). To conclude, the reviewer suggests that this paper should be submitted in a more specialized journal, such as The Journal of Chemical Physics or Biophysical Journal, after adding more deeper 2D neutron diffraction analyses.

In our manuscript we compare previously published experimental pressure–distance curves of PC lipids and DGDG and quantitatively reproduce the differences using state of the art computational/theoretical methods. To this end, we identify the physical mechanisms responsible for the different interaction characteristics of phospholipid and glycolipid membranes.

As pointed out by the two other reviewers, our work uses methodology of unprecedented accuracy and quality and yields important new insights into hydration forces in general as well as into the influence of glycolipids on the interaction of biological membranes in particular.

The results summarized in Fig. 7 correspond to the experimental determination of an important complementary piece of information which was initially missing: The water layer thickness for DGDG at the swelling limit. As mentioned before, this information allows to interpret the simulation results, which constitute the bulk of the present work. Having this in mind, the experimental result cannot be considered "preliminary" but should be considered complementary and conclusive with respect to the particular question addressed. At this point we would like to emphasize that obtaining complementary experimental data when needed for a theoretical interpretation, to our view, is an indication of effective and fruitful collaborations between experimentalists and theoreticians.

As stated earlier, the present study is focused on pressure–distance curves and on their mechanistic interpretation. Off-specular analysis to obtain membrane mechanical properties would go far beyond the scope of this work.

Reviewer #3 (Remarks to the Author):

A. The paper reports a computer simulation study of the interaction between two non-charged lipid bilayers with a disugar head-group and the theoretical study is complemented by experimental measurements. Key findings are

i) The simulations capture the measured interaction profile with an unprecedented accuracy.

ii) Sugar lipids show a clearly more short range repulsion than the corresponding phospholipids (with chains in the liquid state). This has the consequence that there is a stronger attractive minimum for sugar lipids and a larger tendency for bilayer-bilayer cohesion. An affect that the authors argue has physiological relevance.

iii) It is conclude that the nature of the bilayer-bilayer repulsion qualitatively depends on the chemical nature of the head-group. This is in contrast to a wide-spread belief the the nature of the repulsion is dependent on a generic property of the solvent water.

iv) The repulsion found for the sugar lipids is determined by a short-range hydroxyl-water interaction competing with the direct hydroxyl-hydroxyl interaction between sugar moieties.

B. The simulations of the bilayer-bilayer interactions are of unprecedented accuracy and of a quality clearly beyond what other investigators have accomplished. The authors are able to give a detailed picture of the the molecular interactions responsible for the swelling of the bilayer. This is, in my personal view, this has been accomplished for the swelling of electrically neutral amphiphiles. (The phospholipid case has not yet been satisfactorily analysed).

C. The simulation methodology has been used previously by the authors and it is highly appropriate for the problem considered. The data are in general of high quality and they are discussed in a clear way. Some details of the presentation is discussed under point E.

D. The handling of the data is, as far I can judge, fully satisfactory.

E. The main, important conclusions are clearly justified from the data.

We thank the reviewer for this favorable evaluation of our work.

A few details can be improved.

i) The chemical composition of the lipid chains are not clearly specified in the experimental section.

The chain composition in the computer simulations matches the one used in reference by Demé et al. [14] in which the experimental DGDG pressure–distance curve was reported: $\approx 80\%$ dalDGDG and $\approx 20\%$ palDGDG, where dalDGDG has two threefold-unsaturated C18 chains (18:3) and palDGDG has one 18:3 chain at the sn-2 position and one saturated C16 chain (16:0) at the sn-1 position.

This was already explained on page 9 of the original manuscript.

In the revised manuscript (in Methods section, page 12) we also give explicitly the chain composition of the sample experimentally studied at excess hydration (Fig. 7), which also matches the one from reference [14].

ii) It is of some interest to what extent water molecules are mediating the in-plane sugar-sugar interactions. This should be clear fro the simulations.

As suggested by the reviewer, we have carried out an analysis of the extent to which water molecules mediate in-plane sugar-sugar interactions. To this end, we have compared the average number of direct and indirect in-plane hydrogen bonds (HBs, see also points iv and v below) formed

by each glycolipid headgroup. Here, "direct" HBs denote HBs between two sugar headgroups in the same surface, while "indirect" HBs denote HBs between a sugar headgroup and a water molecule that is simultaneously involved in HB with another sugar headgroup in the same surface. Under highly hydrated conditions ($n_w = 30$), the situation that we investigated exemplarily, we find that each sugar headgroup on average is involved in 2.1 ± 0.1 direct HBs and in 1.5 ± 0.1 indirect HBs, indicating that the interactions between the sugar headgroups are water-mediated to a considerable extent. This is now explained on page 4 of the revised manuscript:

"The extent to which water molecules mediate the interaction between saccharide headgroups in the same membrane surface was investigated at high hydration ($n_w = 30$). Each sugar headgroup on average is involved in 2.1 ± 0.1 direct HBs with other headgroups in the same surface and in 1.5 ± 0.1 "indirect" HBs, i.e., HBs with water molecules that are simultaneously involved in an HB with other headgroups in the same surface. This result indicates that the interactions between the saccharide headgroups are water-mediated to a considerable extent."

iii) It could help the reader if the authors specify the ration water molecules per head-group at the swelling limit.

As suggested by the reviewer, we have included the number of water molecules per lipid at the swelling limit, n_w^0 , into the revised manuscript ($n_w^0 \approx 15$ for DGDG, $n_w^0 \approx 30$ – 35 for PC lipids, page 4).

iv) The authors discuss sugar-water interactions referring to both dipole-dipole interactions and to hydrogen bonds. A typical Nature reader isn't aware of the fact that both are electrostatic in nature and closely related. The reader would be helped by a more clear discussion of this point.

According to the reviewer's suggestion, we have included on page 5 of the revised manuscript a sentence pointing out the electrostatic nature of hydrogen bonds:

"On a simplified level, HBs are electrostatic interactions involving two atoms carrying negative partial charges and a hydrogen atom that is covalently bound to one of them and carries a positive partial charge."

v) A related issue is that the authors count hydrogen bonds in a digital manner. It is the firm opinion of the reviewer that this is a futile exercise in a liquid context (except in some special cases)

To quantify the number of hydrogen bonds in the simulations, we employ the widely-used Luzar–Chandler geometric criterion, according to which a HB exists if the distance between donor and acceptor atoms is smaller than 0.35 nm and the hydrogen–donor–acceptor angle is smaller than 30° . Counting HB numbers in this way is reasonable when the angle-distance histogram exhibits a clear maximum for the "bound" state which is well separated from the "un-bound" states. As can be seen for example in Fig. R1, taken from a recent work of Dominik Marx and co-workers using ab initio MD on water at 1 atm and 300 K [Imoto et al. Phys. Chem. Chem. Phys., 17, 24224 (2015)], the distribution has such a well-separated "bound" maximum in the bottom-left corner. The dashed rectangular line corresponds to the Luzar–Chandler criterion, suggesting that HB counting using this criterion, as is commonly done in the literature, is not unreasonable.

In response to the reviewer's comment, we explain the Luzar–Chandler criterion on pages 5/6 of the revised manuscript:

"To extract HB numbers from our simulation trajectories, we employ the widely-used Luzar–Chandler geometric criterion [Luzar & Chandler 1996], according to which a HB is present if the distance between donor and acceptor atoms is smaller than 0.35 nm and the hydrogen–donor–acceptor angle is smaller than 30° "

Figure R1: Angle-distance histogram of water at 1 atm and 300 K obtained in ab initio MD simulations by Imoto et al. [Imoto et al. Phys. Chem. Chem. Phys., 17, 24224 (2015)].

vi) The molecular description of the sugar-water interaction is valid in a much broader context and relevant for a sugar solutions as wells as for glycoproteins. The authors could point this out.

We thank the reviewer for this suggestion. In the revised manuscript we have included the following sentence on page 8:

"Finally, sugar–water interactions are not only relevant for glycolipids but also in a much broader context ranging from sugar solutions [40] and sugar surfactants [41, 42] to glycoproteins and sugar-based biomaterials [43]. Some of the concepts presented here may also apply to those problems."

vii) The authors discuss the zwitterionic phospholipid case as a comparison and trace the difference in terms of the size of the dipoles giving more water polarisation. In the context they refer to the model by Radic& Marçelja seemingly forgetting that this model starts from the situation where surface dipole - water polarisation is gone.

We thank the reviewer for this insightful remark.

We agree that in context with the Marcelja & Radic model the maximum orientation degree at the surfaces is meaningful only together with the decay of the orientation profile into the water layer.

Figure R2 shows a zoom into the water orientation profiles (Fig. 5B) around the center of the water layer for PC lipid and DGDG membranes. Note that the water layer thickness is the same for PC lipids and DGDG ($D_w=2.3$ nm).

It is clearly seen that for PC lipid membranes water orientation is significant essentially all the way to the center of the water slab, where it has to vanish by symmetry. In contrast, for DGDG the water orientation in the water layer is negligible. Figure R2 is now also included in the Supporting Material and referred to in the main text (page 7).

In addition, we have re-structured the discussion of the water orientation profiles, showing water density profiles along with the orientation profiles. The new text now reads:

"Figure 5A presents density profiles of water and lipids in hydrated DGDG and PC lipid membranes at a large separation of $D_w = 2.3$ nm. It is seen that the water density profiles are nearly identical for the two lipid types. In contrast, as shown in Figure 5B, DGDG and PC lipid membranes lead to very different profiles of water orientation perpendicular to the membrane surfaces, $\langle \cos\theta_w \rangle$ (see inset

for the definition of θ_w). For the PC lipids the water dipoles close to the membrane surfaces at $z = \pm D_w/2 = \pm 1.15$ nm (see vertical dashed lines) are strongly oriented and significant orientation extends virtually all the way to the centre of the water layer (see zoom-in in the Supporting Material), where it has to vanish by symmetry. For DGDG, on the other hand, water orientation is pronounced only in the poorly hydrated inner headgroup regions and insignificant inside the water layer."

Figure R2: Zoomed-in region of Fig. 5B in the main text: Water orientation profiles $\langle \cos \theta_w \rangle$ between DGDG and PC lipid membranes at large separation ($D_w = 2.3$ nm).

F. See point E above.

G. The authors seems to have missed that Marra and coworkers have measured interactions between glycolipid bilayers in the mid 1980-ties. (See J. Marra J Colloid Interface Science 1986 and references therein). There is also an expensive literature on the aggregation behaviour of sugar surfactants. These systems show a behaviour in qualitative accordance with the findings of the authors.

We thank the reviewer for reminding us of the important work of Marra. In the revised manuscript (page 2) we have included a sentence in which we refer to it:

"Consistently, surface force apparatus measurements revealed striking differences in the interaction between pure phospholipids and pure glycolipids (MGDG and DGDG) [Marra 1985, Marra 1986]."

In addition, on page 8 we now also state that the sugar–water interactions are relevant for sugar surfactants and included references [41] and [42].

H. The paper is overall clearly written. However, the section on interaction mechanism is difficult to read. This reflects the fundamental problem that a simulation is based on complex relatively strong molecular interactions that collectively produce a relatively weak net bilayer-bilayer interaction. There is no unique way to disentangle the different molecular contributions. The authors make a good effort to solve the problem, but a shortening of this section might make it more clear and possibly also more valid.

We agree with the reviewer that there is no unique way to disentangle the complex interplay of molecular contributions to surface interactions. But instead of shortening the section on interaction mechanisms, which to our opinion could be detrimental to the logical sequence of the text, we decided to include a short opening section in the spirit of the reviewer's comment:

"The interaction of lipid membranes across a water layer involves a complex interplay of competing molecular interactions that collectively produce a relatively weak net repulsion. In the following we analyze our simulations such as to identify the dominant repulsion mechanisms, keeping in mind that alternative ways to disentangle the different molecular contributions clearly exist."

Additional modifications in the manuscript:

Apart from the revisions motivated by the reviewers, we made additional amendments in the manuscript in order to improve the clarity and quality of the paper:

We added Fig. 3B showing the density profiles of the PC lipids, which now enables a direct comparison with the DGDG profiles in Fig. 3A. In order to compare both profiles at the same membrane–membrane separation D_w within our available set of simulations, we also updated Fig. 3A to correspond to the separation of $D_w = 0.23$ nm.

REVIEWERS' COMMENTS:

Reviewer #1 (Remarks to the Author): *ALL (minor) comments made about this beautiful work that will become a "classic" cornerstone in biophysics have been answered to in an excellent way.*

We thank the reviewer for this favourable evaluation of our manuscript.

Reviewer #3 (Remarks to the Author): *The manuscript has gained considerably in clarity by the revision and I recommend publication. There are some qualitative arguments in the paper which I personally don't support. However in these matters I regard the paper as a very valuable contribution to an ongoing scientific debate.*

We thank the reviewer for this favourable evaluation of our manuscript.

Emanuel Schneck
on behalf of all authors